# Revealing Fermi surface evolution and Berry curvature in an ideal type-II Weyl semimetal

Qianni Jiang [1], Johanna C. Palmstrom [2], John Singleton [2], Shalinee Chikara [3], David Graf [3], Chong Wang [4], Yue Shi [4], Paul Malinowski[1], Aaron Wang [1], Zhong Lin[1], Lingnan Shen[1], Xiaodong Xu[1,4], Di Xiao [1,4] & Jiun-Haw Chu [1]✉

In type-II Weyl semimetals (WSMs), the tilting of the Weyl cones leads to the coexistence of electron and hole pockets that touch at the Weyl nodes. These electrons and holes experience the Berry curvature generated by the Weyl nodes, leading to an anomalous Hall effect that is highly sensitive to the Fermi level position. Here we have identified field-induced ferromagnetic $MnBi_{2-x}Sb_xTe_4$ as an ideal type-II WSM with a single pair of Weyl nodes. By employing a combination of quantum oscillations and high-field Hall measurements, we have resolved the evolution of Fermi-surface sections as the Fermi level is tuned across the charge neutrality point, precisely matching the band structure of an ideal type-II WSM. Furthermore, the anomalous Hall conductivity exhibits a heartbeat-like behavior as the Fermi level is tuned across the Weyl nodes, a feature of type-II WSMs that was long predicted by theory. Our work uncovers a large free carrier contribution to the anomalous Hall effect resulting from the unique interplay between the Fermi surface and diverging Berry curvature in magnetic type-II WSMs.

The Weyl semimetal (WSM) is a three-dimensional gapless topological phase characterized by energy bands that intersect at points in momentum space known as Weyl nodes[1–4]. The stability of these band crossings relies on the breaking of either time reversal or inversion symmetry. The low-energy quasiparticles near the Weyl nodes are described by the relativistic Weyl equations, giving rise to Weyl fermions in condensed matter systems. Unlike their high-energy counterpart, the Weyl fermions in WSMs are not constrained by Lorentz symmetry, allowing the Weyl cones to tilt in energy–momentum space[5–9]. When the tilting of the cones is large enough such that the velocity changes sign along the tilting direction, a transition from type-I to type-II WSMs occurs. In a type-I WSM (Fig. 1b), the Fermi surfaces are point-like objects located at the Weyl nodes when the Fermi energy aligns with the Weyl crossing. In contrast, in type-II WSMs (Fig. 1a), the tilting of the cones leads to extended electron and hole Fermi pockets touching the Weyl nodes[10].

The tilting of Weyl cones leads to distinct transport behavior in the two types of WSMs[11–17]. In a type-I WSM without cone tilting, the anomalous Hall effect is solely determined by the location and the topological charge of the Weyl nodes, and the free carriers do not contribute to the anomalous Hall conductivity (AHC) either intrinsically or extrinsically. Consequently, the magnitude of the AHC is independent of the Fermi level position, even when the Fermi level is tuned above or below the Weyl crossings[12,18]. When the Weyl cones are tilted, both Berry curvature and skew scattering effects can contribute to AHC in the presence of Fermi surfaces[19]. Consequently, in a type-II WSM, in addition to the contribution from the Weyl points, the Fermi surfaces give rise to a logarithmically divergent contribution to the AHC, which is only truncated by the finite size of electron and hole pockets[13]. As a result, when the Fermi level is tuned across the type-II Weyl crossing, the rapid changes in the Fermi surface lead to a dramatic variation of the AHC.

[1]Department of Physics, University of Washington, Seattle, WA 98195, USA. [2]National High Magnetic Field Laboratory, Los Alamos National Laboratory, Los Alamos, NM 87545, USA. [3]National High Magnetic Field Laboratory, Florida State University, Tallahassee, FL 32310, USA. [4]Department of Material Science and Engineering, University of Washington, Seattle, WA 98195, USA. ✉e-mail: jhchu@uw.edu

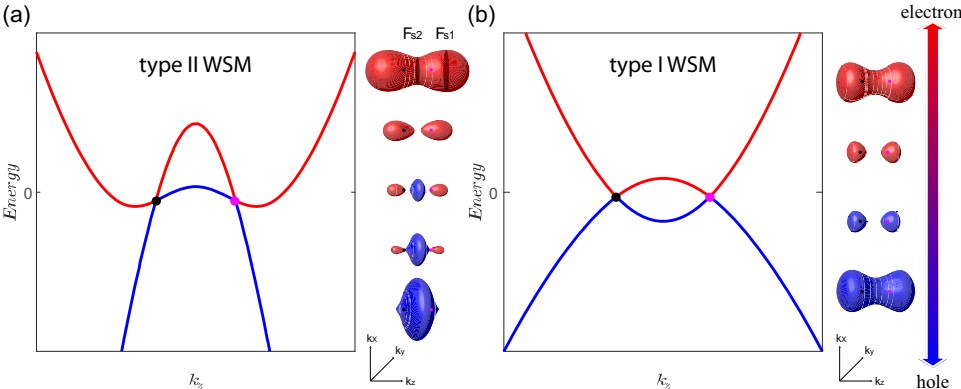

**Fig. 1 | Schematic diagram of band structures and Fermi pockets of an ideal type-II and type-I Weyl semimetals. a** and **b** Band dispersions of ideal type-II (**a**) and type-I (**b**) Weyl semimetals along the $k_z$ direction when a pair of Weyl nodes is located along the Γ−Z cut. On the right, the evolution of the Fermi pockets as the Fermi level moves across the Weyl nodes is exhibited. The red line and pockets refer to electrons, whereas the blue line and pockets represent hole carriers. The magenta and black points represent the Weyl nodes with positive and negative chirality, respectively.

An outstanding challenge in the field of WSMs is to isolate the effects of the Weyl nodes and experimentally observe the theoretically predicted exotic transport effects. Unfortunately, most of the WSMs discovered so far have multiple pairs of Weyl nodes and coexist with other topologically trivial bands, complicating the interpretation of transport measurements. For instance, in $Co_3Sn_2S_2$, a magnetic Weyl semimetal, three pairs of Weyl nodes are observed along with 16 electron pockets and 12 hole pockets[20–23]. Such complex band structures make it very challenging to disentangle the contributions to the AHC from various energy bands and intrinsic/extrinsic mechanisms. A solution to this challenge is to realize a tunable material system with a single pair of Weyl nodes, an ideal WSM, and simultaneously monitor the evolution of the Fermi-surface sections and AHC as the Fermi level is tuned across the Weyl nodes.

In this paper, we provide direct evidence that the field-induced ferromagnetic state of $MnBi_{2-x}Sb_xTe_4$ is the long-sought ideal WSM. By substituting Bi with Sb, we tune the carriers from n-type to p-type, allowing us to study the evolution of the Fermi-surface sections by a combination of quantum oscillations and high-field Hall measurements[24,25]. The observed evolution of the Fermi surfaces is conclusively elucidated by the band structure of a single pair of type-II Weyl nodes. Furthermore, the anomalous Hall conductivity exhibits a strong doping dependence near charge neutrality, displaying a heartbeat-like behavior that is consistent with expectations for a type-II WSM[13] and in agreement with results from density functional theory (DFT) calculations. Our findings provide a definitive identification of an ideal type-II Weyl semimetal in $MnBi_{2-x}Sb_xTe_4$, offering a valuable platform for further studies of Weyl physics.

## Results

$MnBi_2Te_4$ is a van der Waals layered antiferromagnetic topological insulator[26–28]. DFT calculations have predicted that the band structure of $MnBi_2Te_4$ in the field-induced ferromagnetic phase is characterized by a single pair of type-II Weyl nodes[26,29]. However, these calculations have also shown that $MnBi_2Te_4$ can be easily tuned to a type-I WSM or a trivial insulator phase by small amounts of strain[27]. Due to the incompatibility between angle-resolved photoemission spectroscopy (ARPES) and a high magnetic field, it is not possible to directly investigate the band structure of the field-induced state. Here we show that the topological band structure of $MnBi_{2-x}Sb_xTe_4$ can be experimentally determined by the doping dependence of quantum oscillations. As shown in Fig. 1, the electronic structure of individual topological states yields distinct Fermi surface topology as the Fermi level is tuned across the charge neutrality point. The Fermi surface topology can be determined by the number of quantum oscillation frequencies complemented by high-field Hall measurements, serving as a distinctive fingerprint associated with each unique topological state. Previous quantum-oscillation studies of $MnBi_{2-x}Sb_xTe_4$ show that the doping dependence of oscillation frequencies and effective mass is broadly consistent with the DFT calculations[30,31]. Nevertheless, only a single oscillation frequency has been observed, which is insufficient to conclusively determine whether it is a type-I, type-II WSM, or a trivial FM insulator. As we discuss below, the first main result of this work is the observation of extra frequencies that enable us to unambiguously pin down the type-II WSM band structure of $MnBi_{2-x}Sb_xTe_4$.

One of the reasons why it is challenging to resolve multiple frequencies in $MnBi_{2-x}Sb_xTe_4$ is the low oscillation frequency (<100 T) and the limited field range (where oscillations can only be observed above the spin−flip transition at 7 T). To overcome this difficulty, we applied large magnetic fields provided by a 60 T pulsed magnet and a 31 T DC magnet to study a series of $MnBi_{2-x}Sb_xTe_4$ samples with closely spaced chemical dopings near charge neutrality. Figure 2 summarizes representative Shubnikov de-Haas (SdH) oscillation data in $MnBi_{2-x}Sb_xTe_4$. We divided the data into three categories: n-doped ($x < 0.7$, Fig. 2a−f), p-doped ($x > 0.7$, Fig. 2i−l), and near charge neutrality ($x ≈ 0.7$, Fig. 2g, h). All the data were measured at base temperature with the magnetic field along the c-axis. In the case of n-doped samples, the amplitude of the magnetoresistance oscillations shows a dramatic increase when the magnetic field is above 40 T (Fig. 2e, f), whereas for the p-doped samples, the amplitude of oscillations gradually increases with the field, consistent with the damping behavior of single-frequency oscillations (Fig. 2i−l). A closer inspection of the data for the n-doped sample reveals that the change in oscillation amplitude is caused by a beating effect from the presence of multiple frequencies (Fig. 2a−d), which is also observed in data below 40 T (Fig. 2e, f).

Fast Fourier transforms (FFTs) were applied to the oscillatory signals to extract a preliminary overview of the evolution of oscillation frequencies as a function of doping (Fig. 2m). The FFT spectrum reveals that the oscillation frequency decreases as the doping approaches charge neutrality, which is consistent with the previous studies. However, on the n-doped side ($x < 0.7$), two peaks are always observed, whereas on the p-doped side ($x > 0.7$), only one peak is observed. We note that the FFT peaks are broad due to the limited number of observed oscillations, especially for the samples near charge neutrality. In such conditions, a more accurate analysis is obtained by directly fitting the oscillatory signals using the Lifshitz−Kosevich (LK) formula (see details in Supplementary Fig. 2).

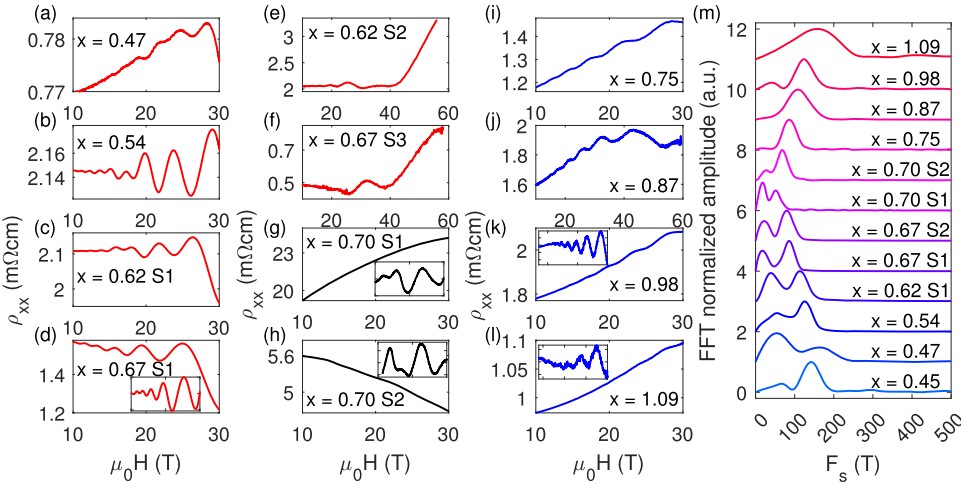

**Fig. 2 | SdH oscillations and FFT of MnBi$_{2-x}$Sb$_x$Te$_4$ under pulsed (60 T) and DC (31 T) magnetic field. a–l** magnetoresistivity of **a–f** electron-doped, **g** and **h** near charge neutrality point, and **i–l** hole-doped MnBi$_{2-x}$Sb$_x$Te$_4$ samples. Insert demonstrates the oscillatory part after polynomial background subtraction. **m** Fast Fourier transform spectrum of the SdH oscillations for various dopings. Each spectrum is offset by 1 for clarity.

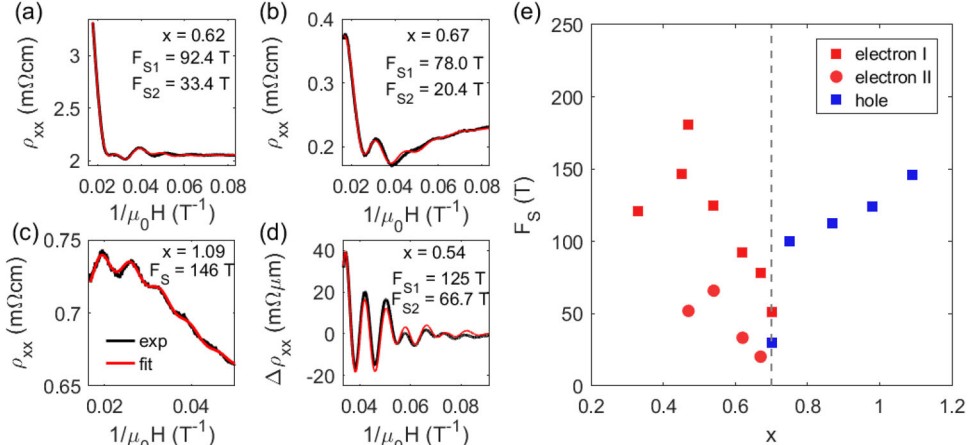

**Fig. 3 | Analytical fitting of SdH oscillations of MnBi$_{2-x}$Sb$_x$Te$_4$.**
**a–c** Magnetoresistivity as a function of $1/\mu_0H$ measured under a 60 T pulsed magnetic field with $\mu_0H$//[001] at 1.5 K and the analytical fit to the Lifshitz–Kosevich (LK) formula for (**a**) electron-doped $x = 0.62$, **b** electron-doped $x = 0.67$, and (**c**) hole-doped $x = 1.09$. **d** Oscillatory part of the magnetoresistivity as a function of $1/\mu_0H$ measured under a 31 T DC magnetic field with $\mu_0H$ //[001] at 1.5 K for electron-doped MnBi$_{2-x}$Sb$_x$Te$_4$ ($x = 0.54$) and the analytical fit to the Lifshitz–Kosevich (LK)

formula. The black lines represent the experimental data, whereas the red lines represent the fit. **e** Doping dependence of oscillation frequencies extracted from the analytical fitting to the LK formula. The red squares denote the band maximum ($k_z \neq 0$) of the electron pocket, whereas the red circles denote the band minimum ($k_z = 0$) of the electron pocket. The blue squares represent the band maximum at $k_z = 0$ of the hole pocket.

Representative examples of the fits are shown in Fig. 3a–d. The fits yield two oscillation frequencies for electron-doped samples and a single frequency of 146 T for p-doped sample with $x = 1.09$. The fit also confirms that the rapid increase of oscillation amplitude above 40 T is indeed caused by a beating effect of two oscillation frequencies as the system approaches the quantum limit. Figure 3e summarizes the frequency as a function of doping extracted from the LK fits, showing good agreement with the FFT analysis.

The oscillation frequency provides us with information about the extremal cross-sectional area of the Fermi surface in the plane perpendicular to the magnetic field, as determined by the Onsager relation ($F_s = \frac{\hbar}{2\pi e}A$). However, it does not indicate whether these frequencies correspond to separate Fermi-surface sections or different extremal cross-sectional areas of a single Fermi surface. To address this question, we turn to the Hall resistivity. Specifically, we focus on the Hall resistivity above the magnetic saturation field, which is

relevant to the electronic structure in the field-induced ferromagnetic state. Our findings, as shown in Fig. 4, reveal that the Hall resistivities are always linear in B, except for the doping closest to charge neutrality ($x \approx 0.7$). A linear Hall resistivity very likely arises from a single type of carrier. Therefore, we conclude that the two frequencies observed in the quantum oscillations arise from the two extremal cross-sectional areas of a single Fermi surface, except for $x = 0.7$.

The high field Hall resistivity of $x = 0.7$ exhibits strong nonlinearity, changing from a positive slope to a negative slope, as shown Fig. 4c. This suggests the coexistence of electron and hole pockets. To extract the electron and hole carrier densities, we fit the Hall resistivity data to the effective two-band model, using the longitudinal resistivity at the upper critical field as a constraint (see Supplementary Fig. 1). The fitting yields an electron density $2.37 \times 10^{17}$ cm$^{-3}$ with a mobility of 1100 cm$^2$/V/s and a hole density $2.35 \times 10^{17}$ cm$^{-3}$ with a mobility of 1300 cm$^2$/V/s. The coexistence of electron and hole carriers rules out

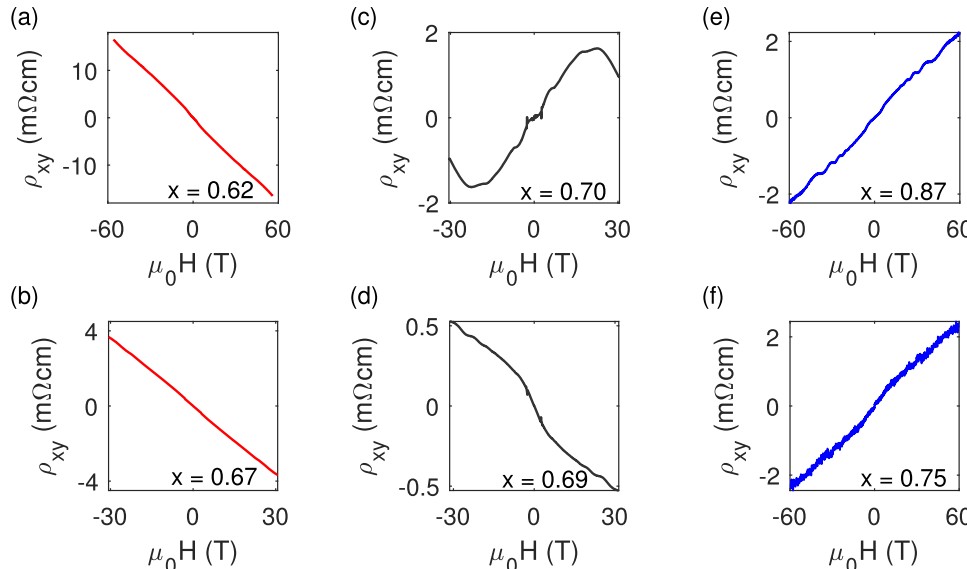

**Fig. 4 | High field Hall resistivity of MnBi$_{2-x}$Sb$_x$Te$_4$. a–f** Hall resistivity with $\mu_0$**H**//[001] at 1.5 K for electron-doped (**a**) $x = 0.62$, **b** $x = 0.67$, near the charge neutrality point (**c**) $x = 0.70$, **d** $x = 0.69$, and hole-doped (**e**) $x = 0.87$, **f** $x = 0.75$ samples.

the possibility of MBST being a type-I WSM or gapped insulator, as neither scenario allows for the coexistence of electron and hole pockets.

We can now compare the observed quantum-oscillation frequencies with the predicted Fermi surfaces based on the type-II WSM band structure (Fig. 1a). When the Fermi level is slightly above the Weyl nodes and the Lifshitz energy, a single electron pocket encloses two Weyl points and has a dumbbell shape aligned with the $\mathbf{k}_z$ axis where the two Weyl nodes are located. As a result, there are three extremal orbits - one minimum at $\mathbf{k}_z = 0$ and two equivalent maxima at $\mathbf{k}_z \neq 0$. The minimum and maxima correspond to the two frequencies observed in slightly n-doped samples ($0.47 < x < 0.7$). When the Fermi level is well below the Weyl nodes, a single hole pocket with a single extremal orbit at $\mathbf{k}_z = 0$ is expected, which agrees with the single oscillation frequency observed in hole-doped samples ($x > 0.7$). Finally, when the Fermi level is below the Lifshitz energy and near the Weyl points, two identical electron pockets and a hole pocket are expected, each with a single extremal orbit. This explains the two frequencies observed in the $x = 0.7$ sample. Remarkably, the carrier density calculated from the oscillation frequency, assuming spherical Fermi surfaces, qualitatively agrees with carrier density extracted from the two-band fitting of the Hall resistivity. This agreement suggests that the quantum oscillation measurements have captured all the carriers participating in the transport, and the Fermi surface evolution described above provides a complete characterization of the electronic structure of this system.

After identifying the Fermi surface topologies of MnBi$_{2-x}$Sb$_x$Te$_4$, we can now investigate the unique transport signature associated with an ideal type-II WSM. A well-known result of the TRS breaking WSM is that the AHC is a topological quantity determined by the separation of Weyl nodes in momentum space, independent of the position of the Fermi level. However, this is only the case for a type-I WSM without cone tilting, where the free carriers do not contribute to the AHC. The reason is that the integration of the z-component of Berry curvature ($\Omega_Z$) over all occupied states (for electrons) or unoccupied states (for holes) is completely canceled, giving rise to zero free carrier contribution to AHC (Fig. 5a). In contrast, in a type-II WSM the electron pockets capture only positive $\Omega_Z$ and the single hole pocket captures only negative $\Omega_Z$, which aggregates a large free carrier contribution to AHC (Fig. 5b). When the Fermi level is tuned across the Weyl nodes and

the electron and hole pockets shrink or emerge, the integration of Berry curvature changes sign. As a result, the AHC of a type-II Weyl semimetal is expected to exhibit a heartbeat-like behavior as a function of doping. To explore this effect, we conducted a detailed study of the AHC as a function of doping near the charge neutrality point.

Figure 5c displays the field dependence of the anomalous Hall resistivity for a series of chemical dopings obtained by subtracting the ordinary Hall resistivity background (see details in Supplementary Fig. 3). The inset of Fig. 5c summarizes the magnitude of the anomalous Hall resistivity as a function of doping. With increasing Sb concentration, the anomalous Hall resistivity changes sign from negative to positive and reaches a maximum magnitude at the charge neutrality point ($x = 0.7$). We then converted the anomalous Hall resistivity to AHC and plotted it as a function of carrier density in the AFM state extracted from the low-field Hall resistivity (Fig. 5d). This carrier density is considered as a proxy of the Fermi level position. Remarkably, a singular AHC with an upturn followed by a downturn was observed as the Fermi level is tuned from above to below the Weyl points. This behavior is in excellent agreement with the expectations for an ideal type-II WSM as discussed above. It is also qualitatively consistent with the DFT calculations (see inset of Fig. 5d). However, a quantitative discrepancy is observed between the experimental and DFT-calculated AHC values, which is probably to do with the detail differences in the calculated and real band structures[32] and extrinsic contributions to the AHC. Nevertheless, the heartbeat pattern of the AHC observed here is a characteristic feature of an ideal type II Weyl semimetal, independent of details in band structures. This observation further confirms that MnBi$_{2-x}$Sb$_x$Te$_4$ is indeed an ideal type-II WSM.

In conclusion, our findings provide direct evidence for the field-induced ferromagnetic MnBi$_{2-x}$Sb$_x$Te$_4$ being an ideal type-II Weyl semimetal, with a single pair of Weyl nodes near the Fermi energy. Through the combined techniques of quantum oscillations and high-field Hall measurements, we demonstrate that the evolution of the Fermi surface can only be explained by the band structure of an ideal type-II Weyl semimetal. Furthermore, we observed a strong doping dependence of the anomalous Hall conductivity near charge neutrality, exhibiting a heartbeat-like behavior consistent with expectations for a type-II WSM and supported by DFT calculations. Our work highlights MnBi$_{2-x}$Sb$_x$Te$_4$ as a promising platform for further investigations into Weyl physics. Moreover, the highly tunable nature of MnBi$_{2-x}$Sb$_x$Te$_4$ with an electronic

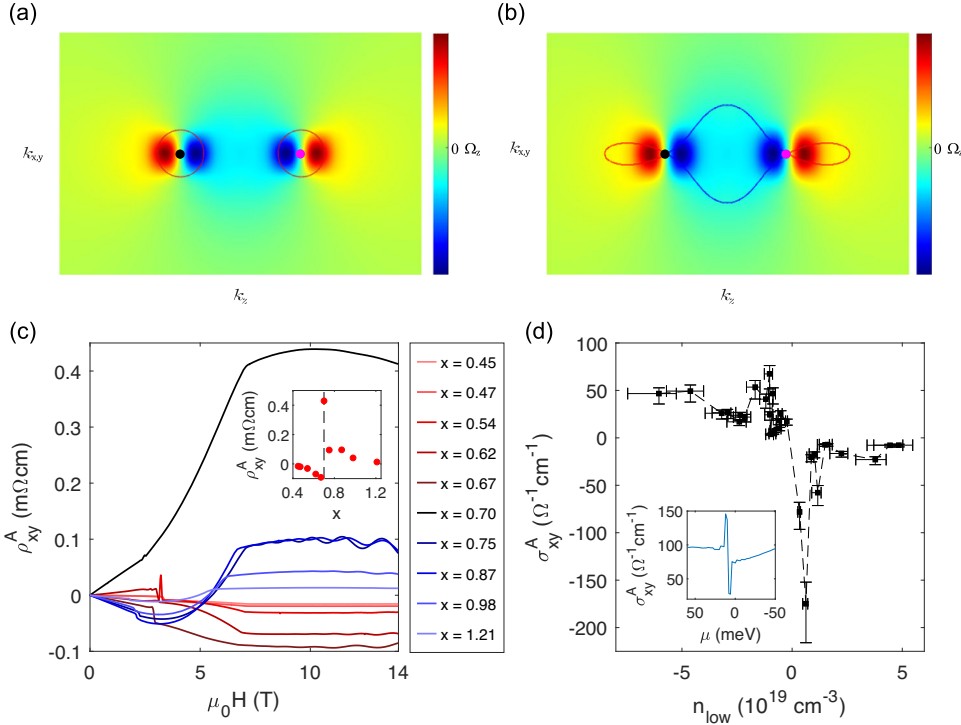

**Fig. 5 | The evolution of Berry curvature and anomalous Hall effect near the Weyl points. a** and **b** $Z$-component of Berry Curvature near a pair of Weyl nodes along the $k_z$ direction as a function of $k_z$, and $k_{x,y}$. The red and blue lines denote the electron and hole pockets close to the Weyl nodes of (**a**) a type-I Weyl semimetal and (**b**) a type-II Weyl semimetal. The magenta and black points represent the Weyl nodes with positive and negative chirality, respectively. **c** Anomalous Hall resistivity (AHR) as a function of magnetic field strength with $\mu_0\mathbf{H}$//[001] at 2 K for different chemical dopings. Inset of (**c**) shows the doping dependence of AHR. **d** Anomalous Hall conductivity (AHC) as a function of carrier de*n*sity ($n_{low}$) extracted by fitting the low-field Hall resistivity linearly. The error bar represents the standard deviation of the experimental data caused by the uncertainty in the measurements of sample dimensions. Insert of (**d**) demonstrates the DFT calculations of the AHC as a function of chemical potential in field-induced FM MnBi$_2$Te$_4$.

structure sensitive to external fields, such as strain, pressure, chemical doping, and magnetic field angle, makes it a highly versatile platform for exploring topological phase transitions[26,27,33–35].

## Methods
Single crystals of MnBi$_{2-x}$Sb$_x$Te$_4$ were grown using a Bi (Sb)–Te self-flux[25,36]. The compositions of the crystals were determined by conducting elemental analysis on a cleaved surface of multiple samples within a batch, using a Sirion XL30 scanning electron microscope. The high field magnetotransport measurements were carried out at the National High Magnetic Field Laboratory in Los Alamos (using a 65 T pulsed magnet) and Tallahassee (using a 31 T resistive DC magnet). The anomalous Hall measurements were conducted in a 14 T Physical Property Measurement System (PPMS) at the University of Washington. The magnetotransport measurements were performed using a standard four-probe or six-probe contact configuration, with the current direction in-plane and the magnetic field out-of-plane (*c*-axis). To eliminate potential effects from contact misalignment, the magnetoresistivities and Hall resistivities were symmetrized and anti-symmetrized, respectively.

Density functional theory calculations (DFT) were performed by Vienna Ab initio Simulation Package (VASP)[37,38], with a modified Becke–Johnson (mBJ) functional[39] for electron exchange and correlation potentials. We employed the projector augmented wave (PAW)[40] method for electron–ion interaction and an energy cutoff of 269.9 eV for wavefunction. A $\Gamma$ centered $12 \times 12 \times 4$ $k$-point sampling was used. Maximally localized Wannier functions were generated using the WANNIER90 software package[41]. The anomalous Hall conductivity calculation was conducted using the Kubo formula with the Wannier interpolation scheme[42].

## Data availability
All data needed to evaluate the conclusions are present in the paper and supplementary materials. Additional data are available from the corresponding author upon reasonable request.

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

## Acknowledgements

This work was solely supported as part of Programmable Quantum Materials, an Energy Frontier Research Center funded by the U.S. Department of Energy (DOE), Office of Science, Basic Energy Sciences (BES), under award no. DE-SC0019443. A portion of this work was performed at the National High Magnetic Field Laboratory, which is supported by National Science Foundation Cooperative Agreement No. DMR-1644779 and the State of Florida. J.S. and J.C.P. acknowledge support from the DOE BES program "Science at 100 T". J.S. acknowledges the provision of a Visiting Professorship at Oxford University. Both opportunities permitted the design and construction of some of the specialized equipment used in the pulsed-field studies.

## Author contributions

J.-H.C. and Q.J. proposed and designed the research. Q.J. carried out the magnetotransport measurements with the help of J.C.P., J.S. and P.M. under a 65 T pulsed magnetic field and S.C., D.G. and Y.S. under a 31 T DC magnetic field. Single crystals of $MnBi_{2-x}Sb_xTe_4$ were synthesized by Q.J. with the assistance of Y.S., Z.L. and A.W. The evolution of Fermi surface cross-section and anomalous Hall conductivity data were analyzed by J.-H.C. and Q.J. with the help of X.X., C.W. and D.X. Theoretical calculations were carried out by D.X., C.W. and L.S. Q.J. and J.-H.C. wrote the paper with input from all co-authors. J.-H.C. oversaw the project.

## Competing interests

The authors declare no competing interests.
