## [Peer Review File · Nature Communications]

Reviewers' comments:

Reviewer #1 (Remarks to the Author):

See the report in Review Attachments.

Reviewer #2 (Remarks to the Author):

MnBi₂-xSbxTe₄ has been studied in the literature as a promising candidate for an ideal Weyl semimetal (ref. 29). The authors performed very systematic studies on MnBi₂-xSbxTe₄ single crystals by combining DFT calculations and high-field magneto-transport measurements. They concluded that they found strong evidence for the ideal Weyl fermion close to the Fermi energy. However, I do not find direct Weyl point signatures. There is no spectroscopic evidence for Weyl points and Fermi arcs as signatures of Weyl semimetals. First-principles calculations have been done to confirm that Weyl points are located near the Fermi energy level. I do not quite believe that they achieved their goal of pinning down MnBi₂-xSbxTe₄ as an ideal Weyl semimetal. The claim of the ideal Weyl semimetallic nature of the material depends mainly on the theoretical calculations and quantum oscillations, which makes the advancement made by the manuscript at this point rather incremental than a breakthrough. Similar studies have already been done on MnBi₂-xSbxTe₄ (Ref. 28 and 29). How do the present results distinguish from the previous findings? I do not see the stand-out point in the current paper. For this reason, I do not believe that the current work merits the publication in a high impact journal such as Nature Communications. I recommend that the manuscript be submitted to a more specialized journal.

Reviewer #3 (Remarks to the Author):

The manuscript by Qianni Jiang et al. presents the systemic research about the fermi surface evolution in the ideal type-II weyl semimetal, Sb-doped MnBi₂Te₄ system, by analyzing the quantum oscillations and anomalous Hall effect in samples with different doping level which shifted the fermi level crossing the charge neutrality point. The results consider that MnBi₂-xSbxTe₄ as a promising platform for the type-II weyl semimetal.

The manuscript is well written and the research is systematic. Overall the manuscript is publishable in Nature Communications. Nevertheless, some clarifications are in need on the following points.

1. The authors described a clear and concise image about the Fermi surface evolution with the carrier filling crossing the charge neutrality point. The sample can be divided into three categories: n-doped ($x < 0.7$), p-doped ($x > 0.7$), and near charge neutrality ($x \approx 0.7$). In the sample $x \approx 0.7$, two different oscillations have been observed. How to determine the electron and hole of the two pockets?

2. In the sample $x \approx 0.7$, the authors considered that the two frequencies observed in the quantum oscillations arise from an electron pocket and a hole pocket. What about the topological characteristics of the two pockets? Considering the authors have analyzed the quantum oscillations by fitting the LK formula, what is the Berry phase of the two frequencies? Moreover, it will be more convincing if the topological property of the quantum oscillations can be cleared in other samples.

3. It is usually considered that the AHE will be more clear when the Weyl point is closer to the Fermi level. Generally, the carrier density will be in the low level. However, there is some bias between the carrier density and the maximum value of the AHE. What does it come from? Can it be explained by the extrinsic contributions?

If the critiques listed above can be addressed, I believe this manuscript is suitable for publication in Nature Communications.

In the paper, the authors experimentally studied the Fermi surface evolution through the SdH oscillation and high-field Hall measurements, providing evidence for the existence of ideal type-II Weyl points in $\text{MnBi}_{2-x}\text{Sb}_x\text{Te}_4$. They also measured the anomalous Hall conductivity (AHC), finding a heartbeat-like behavior as the Fermi level is tuned across the Weyl point. This behavior is in stark contrast to the type-I Weyl semimetal where the AHC remains almost unchanged during this process. The heartbeat-like behavior provides further evidence for the presence of type-II Weyl points in $\text{MnBi}_{2-x}\text{Sb}_x\text{Te}_4$ since the feature is unique to the type-II Weyl semimetal. While reading the paper, I find it very interesting and well written and provide strong evidence to establish $\text{MnBi}_{2-x}\text{Sb}_x\text{Te}_4$ as an ideal type-II Weyl semimetal. Before I can make the decision to recommend the paper for publication in Nature Communications, I also have some questions and comments and hope that the authors can address.

1. I would like to point out that the type-II Weyl points have also been independently proposed in another work (PRL 115, 265304 (2015)), where they are dubbed structured Weyl points. In fact, it becomes earlier than the Nature paper (your Ref. [5]). There, the ideal type-II Weyl points with only one pair of Weyl points are studied. I thus suggest the authors include the paper in the reference list.
2. In lines 149-150, the authors state that “The reason is that the Fermi surfaces are perfect spheres enclosing the Weyl points for Weyl cones with no tilting”. It is very hard to have a perfect sphere for the Fermi surface even in the type-I case without tilting. For example, for the Weyl Hamiltonian, $H = v_x k_x \sigma_x + v_y k_y \sigma_y + v_z k_z \sigma_z$, if v_x , v_y and v_z are not equal, then the Fermi surface is not perfect sphere. I were wondering whether the perfect spherical Fermi surfaces are required to cancel the contribution of Berry curvature. It looks that Ref. [11] does not have this requirement.
3. Previously, type-II Weyl points have been identified through angle-resolved photoemission spectroscopy, e.g., Ref.[8]. In light of the relatively simple structure of Weyl points in this material, I am confused about why the ARPES cannot be used in this material. Could the authors comment on this?
4. It looks that the experimental result in Fig. 5(d) cannot be simply obtained by shifting the theoretical one by 100. I suggest the authors modify the description.

Reviewer 1

In the paper, the authors experimentally studied the Fermi surface evolution through the SdH oscillation and high-field Hall measurements, providing evidence for the existence of ideal type-II Weyl points in $\text{MnBi}_{2-x}\text{Sb}_x\text{Te}_4$. They also measured the anomalous Hall conductivity (AHC), finding a heartbeat-like behavior as the Fermi level is tuned across the Weyl point. This behavior is in stark contrast to the type-I Weyl semimetal where the AHC remains almost unchanged during this process. The heartbeat-like behavior provides further evidence for the presence of type-II Weyl points in $\text{MnBi}_{2-x}\text{Sb}_x\text{Te}_4$ since the feature is unique to the type-II Weyl semimetal. While reading the paper, I find it very interesting and well written and provide strong evidence to establish $\text{MnBi}_{2-x}\text{Sb}_x\text{Te}_4$ as an ideal type-II Weyl semimetal. Before I can make the decision to recommend the paper for publication in Nature Communications, I also have some questions and comments and hope that the authors can address.

We thank the reviewer's careful reading and are delighted that they find our study "very interesting and well written and provide strong evidence to establish $\text{MnBi}_{2-x}\text{Sb}_x\text{Te}_4$ as an ideal type-II Weyl semimetal."

1. I would like to point out that the type-II Weyl points have also been independently proposed in another work (PRL 115, 265304 (2015)), where they are dubbed structured Weyl points. In fact, it becomes earlier than the Nature paper (your Ref. [5]). There, the ideal type-II Weyl points with only one pair of Weyl points are studied. I thus suggest the authors include the paper in the reference list.

We appreciate the reviewer for pointing out another important theoretical work on type-II Weyl semimetals. We've added PRL 115, 265304 (2015) to our reference.

2. In lines 149-150, the authors state that "The reason is that the Fermi surfaces are perfect spheres enclosing the Weyl points for Weyl cones with no tilting". It is very hard to have a perfect sphere for the Fermi surface even in the type-I case without tilting. For example, for the Weyl Hamiltonian, $H = v_x k_x \sigma_x + v_y k_y \sigma_y + v_z k_z \sigma_z$, if v_x , v_y and v_z are not equal, then the Fermi surface is not perfect sphere. I were wondering whether the perfect spherical Fermi surfaces are required to cancel the contribution of Berry curvature. It looks that Ref. [11] does not have this requirement.

We agree that spherical Fermi pocket is a sufficient but not a necessary requirement for the full cancellation of the Fermi surface contribution. In principle, any shapes of Fermi pockets that enclose net zero z-component of Berry curvature should have zero Fermi surface contribution to AHC. In the context of an ideal type-I Weyl semimetal, the tilting of the Weyl cones does give rise to a Fermi surface contribution to AHC due to uncompensated Berry curvature. Nevertheless, this contribution is small unless the degree of tilting approaches the critical value of the type-I to type-II transition. See Fig. 3 of *Jetp Lett.* 103, 717–722 (2016) (ref. 13 in the revised manuscript). We've revised this sentence to avoid misunderstanding.

3. Previously, type-II Weyl points have been identified through angle-resolved photoemission spectroscopy, e.g., Ref.[8]. In light of the relatively simple structure of Weyl points in this material, I am confused about why the ARPES cannot be used in this material. Could the authors comment on this?

Indeed, ARPES is a powerful probe for identifying Weyl semimetals. However, it is not compatible with strong magnetic fields. In this case, the ideal Weyl semimetal phase resides in the field-induced FM state ($>7T$), making SdH oscillations and AHC measurements the most proper and accessible probe. We clarified this point in our revised manuscript.

4. It looks that the experimental result in Fig. 5(d) cannot be simply obtained by shifting the theoretical one by 100. I suggest the authors modify the description.

We thank the reviewer for pointing out the problem. We've modified the description as reviewer suggested.

Reviewer 2

MnBi_{2-x}Sb_xTe₄ has been studied in the literature as a promising candidate for an ideal Weyl semimetal (ref. 29). The authors performed very systematic studies on MnBi_{2-x}Sb_xTe₄ single crystals by combining DFT calculations and high-field magneto-transport measurements. They concluded that they found strong evidence for the ideal Weyl fermion close to the Fermi energy. However, I do not find direct Weyl point signatures. There is no spectroscopic evidence for Weyl points and Fermi arcs as signatures of Weyl semimetals. First-principles calculations have been done to confirm that Weyl points are located near the Fermi energy level. I do not quite believe that they achieved their goal of pinning down MnBi_{2-x}Sb_xTe₄ as an ideal Weyl semimetal. The claim of the ideal Weyl semimetallic nature of the material depends mainly on the theoretical calculations and quantum oscillations, which makes the advancement made by the manuscript at this point rather incremental than a breakthrough. Similar studies have already been done on MnBi_{2-x}Sb_xTe₄ (Ref. 28 and 29). How do the present results distinguish from the previous findings? I do not see the stand-out point in the current paper. For this reason, I do not believe that the current work merits the publication in a high impact journal such as Nature Communications. I recommend that the manuscript be submitted to a more specialized journal.

We respectfully disagree with reviewer 2. One of the main achievements of this work is to pin down MnBi_{2-x}Sb_xTe₄ as an ideal type-II Weyl semimetal by quantum oscillations and high field Hall effect WITHOUT the need of DFT calculations or any other measurements, which was not achieved in Ref. 28 and 29.

As discussed in the introduction of our manuscript, the electronic structures of type I Weyl semimetal, type II Weyl semimetal and trivial semiconductor would result in different Fermi surface topology as the Fermi level is tuned across the charge neutrality point. The Fermi surface topology can be determined by the numbers of frequencies in quantum oscillations at different dopings, which acts as a unique fingerprint to distinguish between different topological states. In ref 28 and 29, it was impossible to do that because only one quantum oscillation frequency was resolved in all dopings, hence no definite conclusion can be made. In ref. 28 (our previous work) we made a comparison with DFT calculations, whereas ref. 29 relied on the interpretation of magneto-transport measurements, where it claimed the observation of "an intrinsic anomalous Hall effect and negative c-axis longitudinal magnetoresistance attributable to the chiral anomaly," it is important to recognize that neither of these two effects are unique to type-II WSM. The intrinsic AHC exists in any system with non-zero Berry curvature, and the negative magnetoresistance is very common in magnetic materials undergo field induced metamagnetic transition.

In contrast, we observed extra quantum frequencies that were not observed in ref. 28 and 29 under a magnetic field up to 60T on high quality samples with fine dopings. The doping evolution of these frequencies uniquely correspond to the Fermi surface topology of type-II Weyl semimetal phase, allowing us to unambiguously pin down the topological ground state of this material. We believe that this is an important breakthrough because the Weyl semimetal state of $\text{MnBi}_{2-x}\text{Sb}_x\text{Te}_4$ requires an external magnetic field to induce ferromagnetism, making it impossible for ARPES measurement. While STM is compatible with magnetic field, it cannot resolve the Fermi arcs in this case because the separation of Weyl points is along the c-axis, which is perpendicular to the surface layer where STM measures. In other words, it is impossible to use spectroscopy method to pin down the Weyl semimetal phase of $\text{MnBi}_{2-x}\text{Sb}_x\text{Te}_4$. In this sense, our work not only established a new method to probe the electronic structure of topological materials, but it might also be the only way to experimentally pin down the topological phase of a field induced ferromagnet.

In addition to experimentally determining the electronic structure of type-II Weyl semimetal phase in $\text{MnBi}_{2-x}\text{Sb}_x\text{Te}_4$, we also observed a striking heartbeat pattern in the doping dependence of anomalous Hall effect. The strong doping dependence of AHE in type-II Weyl semimetal was discussed in several theory works but was never observed experimentally. We provided a clear picture of why the free carrier contribution of AHE has strong doping dependence by connecting the evolution of Fermi surface and the Berry curvature near a type-II Weyl points.

Based on the reasons stated above, we believe that our work represents a significant breakthrough in the study of magnetic topological materials, and it is ideally suited for publication in Nature Communications. In the revised manuscript we have explicitly emphasized the novelty of our work.

Reviewer 3

The manuscript by Qianni Jiang et al. presents the systemic research about the fermi surface evolution in the ideal type-II weyl semimetal, Sb-doped MnBi_2Te_4 system, by analyzing the quantum oscillations and anomalous Hall effect in samples with different doping level which shifted the fermi level crossing the charge neutrality point. The results consider that $\text{MnBi}_{2-x}\text{Sb}_x\text{Te}_4$ as a promising platform for the type-II weyl semimetal.

The manuscript is well written and the research is systematic. Overall the manuscript is publishable in Nature Communications. Nevertheless, some clarifications are in need on the following points.

We appreciate this reviewer's positive comments and recommendation for publication. We addressed the three scientific questions below.

1. The authors described a clear and concise image about the Fermi surface evolution with the carrier filling crossing the charge neutrality point. The sample can be divided into three categories: n-doped ($x < 0.7$), p-doped ($x > 0.7$), and near charge neutrality ($x \approx 0.7$). In the sample $x \approx 0.7$, two different oscillations have been observed. How to determine the electron and hole of the two pockets?

The carrier type of the two pockets in the $x=0.7$ case is determined by comparing the carrier densities calculated based on quantum oscillation frequencies and that based on the two-band Hall fitting. In the

two-band Hall fitting, the electron carrier density is always larger than the carrier density of the hole band, regardless of the constraint we used for the fitting. Therefore, we attribute the pocket with a greater oscillation frequency to electron pockets (larger Fermi pocket cross section and a greater carrier density, assuming similar Fermi surface anisotropy for both pockets).

2. In the sample $x \approx 0.7$, the authors considered that the two frequencies observed in the quantum oscillations arise from an electron pocket and a hole pocket. What about the topological characteristics of the two pockets? Considering the authors have analyzed the quantum oscillations by fitting the LK formula, what is the Berry phase of the two frequencies? Moreover, it will be more convincingness if the topological property of the quantum oscillations can be cleared in other samples.

We thank the reviewer's comment. As pointed out in a recent theoretical study (PRX 8, 011027 (2018)), the Berry phase of quantum oscillations is only quantized for certain space group symmetry and oscillation orbits. Unfortunately, $\text{MnBi}_{2-x}\text{Sb}_x\text{Te}_4$ in the field induced ferromagnetic phase does not belong to these space group symmetries, hence the phase of oscillation cannot be used to determine band topology. This is discussed in our previous work, PRB 103 (20), 205111 (ref. 30 in the revised manuscript)

3. It is usually considered that the AHE will be more clear when the Weyl point is closer to the Fermi level. Generally, the carrier density will be in the low level. However, there is some bias between the carrier density and the maximum value of the AHE. What does it come from? Can it be explained by the extrinsic contributions?

It is generally true that in a type-I Weyl semimetal when the Weyl point is closer to the Fermi level, the Fermi surface volume shrinks to zero and the AHE is dominated by the topological term. However, in the case of a type-II Weyl semimetal, there are always electron and hole Fermi pockets when the Fermi level is close to the Weyl points. These Fermi pockets pick up the Berry curvature generated by the Weyl points, giving rise to the so called "free carrier contribution" with a heartbeat-like doping dependence, as shown Fig.5 (b). When the Fermi level is slightly above (below) the Weyl points, the integration of z-component of Berry curvature in the Fermi surfaces (i.e. Fermi surface contribution to the AHC) is positive (negative) and maximized (minimized). Therefore, even just considering the intrinsic Berry curvature effect we do not expect a maximum AHC when Fermi level is exactly at charge neutrality. See Eq. 9 in *Jetp Lett.* 103, 717–722 (2016) (ref. 13 in the revised manuscript).

To our knowledge, there is no theoretical study of the extrinsic contributions of AHC in a type-II Weyl semimetal. Nevertheless, given the qualitative agreement of the hear-beat like feature between experiment and the theory calculation that only consider intrinsic contribution, we infer that the extrinsic contribution plays a minor role in the strong doping dependence of AHC.

REVIEWERS' COMMENTS

Reviewer #1 (Remarks to the Author):

The authors have addressed all my questions and comments. Compared to Refs. [28,29], they found extra quantum frequencies and heartbeat pattern that are unique to type-II Weyl semimetals. I thus think that the work represents a significant breakthrough in identifying $\text{MnBi}_{2-x}\text{Sb}_x\text{Te}_4$ as a type-II Weyl semimetal and should be publishable in Nature Communications.

Reviewer #2 (Remarks to the Author):

The authors have addressed all my concerns and justified the novelty, and have established the ideal Weyl semimetallic nature of $\text{MnBi}_{2-x}\text{Sb}_x\text{Te}_4$ by high-field magneto-transport measurements. Therefore, I recommend publication of this paper in Nature Communications.

Reviewer #3 (Remarks to the Author):

This manuscript presents evidence of the type-II Weyl semimetal in the Sb-doped MnBi_2Te_4 system by systemic study of the transport property measurements, which is usually difficult for ARPES experiments in this magnetic system. Considering the systematic research and persuasive analysis, I support this manuscript is publishable in Nature Communications.